# Antimicrobial Effect of Copper Nanoparticles on Relevant Supragingival Oral Bacteria

**DOI:** 10.3390/microorganisms12030624

**Published:** 2024-03-20

**Authors:** Nia Oetiker, Daniela Salinas, Joaquín Lucero-Mora, Rocío Orellana, Mariana Quiroz-Muñoz, Denisse Bravo, José M. Pérez-Donoso

**Affiliations:** 1BioNanotechnology and Microbiology Laboratory, Center for Bioinformatics and Integrative Biology, Facultad de Ciencias de la Vida, Universidad Andrés Bello, Santiago 8370133, Chile; nia.oetiker.mancilla@gmail.com; 2Oral Microbiology Laboratory, Dentistry Faculty, Universidad de Chile, Santiago 8330015, Chile; daniela.salinas.diaz@gmail.com; 3Laboratory of Periodontal Biology, Faculty of Dentistry, Universidad de Chile, Santiago 8330015, Chile; joaquin.lucero.mora@gmail.com; 4Laboratory of Oral Biology, Dentistry Faculty, Universidad Finis Terrae, Santiago 7501015, Chile; 5Scanning Electron Microscopy Laboratory, Faculty of Dentistry, Universidad de Chile, Santiago 8330015, Chile; rorellana@odontologia.uchile.cl; 6School of Medicine, Faculty of Medical Sciences, Universidad de Santiago de Chile, Santiago 9170022, Chile; mquirozu@uc.cl; 7Laboratory of Microbial Interactions, Faculty of Dentistry, Universidad Andrés Bello, Santiago 8370133, Chile

**Keywords:** copper, nanoparticles, biofilm, caries, *Streptococcus*, *Lactobacillus*

## Abstract

Copper nanoparticles (Cu NPs) show promise in dentistry for combating bacterial dysbiosis and tooth decay. Understanding their effects on commensal versus pathogenic bacteria is vital for maintaining oral health balance. While Cu NPs demonstrate antibacterial properties against various oral bacteria, including common pathogens associated with tooth decay, their impact on commensal bacteria requires careful examination. In our work, we analyzed three types of Cu NPs for their effects on the growth, viability, and biofilm formation of representative caries-associated and commensal oral bacteria. *S. sanguinis* showed high tolerance to all Cu NPs, while *L. rhamnosus* was highly sensitive. Oxide-Cu NPs exhibited a stronger inhibitory effect on pathobionts compared with commensal bacteria. Moreover, the biofilm formation of the key cariogenic bacteria *S. mutans* was reduced, with minimal negative effects on commensal species’ biofilm formation. All our results showed that CuO nanoparticles (CuO NPs) exhibit reduced toxicity toward commensal bacteria growth and development but have a strong impact on pathogens. This suggests their potential for targeted treatments against pathogenic bacteria, which could help in maintaining the balance of the oral bacterial community.

## 1. Background

Tooth decay disease represents a major health concern worldwide, resulting in high treatment costs. Globally, it is estimated that more than 2 billion people suffer from caries in permanent teeth, and 520 million children have caries in primary teeth [1].

The oral bacterial community presents a diverse composition including more than 700 bacterial species [2], with most of them playing important roles in oral health maintenance (eubiosis) [3]. Certain interactions are beneficial, such as those that impair the growth of pathobiont bacteria [4]. When the characteristics of the oral cavity favor the presence of pathobiont bacteria, a shift in the microbiota composition occurs toward an increased proportion of acidogenic and aciduric bacteria, resulting in dysbiosis in the oral cavity, therefore promoting the onset and progression of dental caries [5]. In this context, the exposure of dental biofilms to dietary sugars leads to their fermentation into organic acids, resulting in the greater presence of acidogenic and aciduric species. *S. mutans* is particularly recognized for its ability to adhere to tooth surfaces, establishing biofilms that create an optimal environment for bacterial growth and acid production, consequently contributing to the demineralization process [6]. Within this acidic environment, other acidogenic species, such as *L. rhamnosus*, can also thrive and form biofilms [7] and, hence, with acidogenic bacteria predominating, decrease the proliferation and biofilm formation of other species associated with oral health, such as *S. sanguinis* [8].

The colonization and growth of cariogenic bacteria can be limited by commensal bacteria that adhere to tooth surfaces and grow substantially better than pathobionts and many other aciduric species in the absence of sucrose [4]; e.g., *S. sanguinis* is a biological antagonist that represses *S. mutans* by H_2_O_2_ production [9]. Additionally, *S. salivarius* can secrete bacteriocins, inhibiting *S. mutans* [10,11] and promoting a healthy oral state.

When commensal bacteria fail to effectively control the growth of pathogens, chemical treatments with anticavity oral agents (AOAs) become necessary. However, conventional methods like chlorhexidine mouthwashes and antibiotics have drawbacks, as they target a broad spectrum of bacteria, impacting the entire bacterial community indiscriminately [12,13,14]. Therefore, there is an urgent need to develop new AOAs that promote eubiosis without disrupting the balance of the oral microbiome.

In this context, different types of nanoparticles (NPs) have been applied in dental materials [15] (FeOx, ZrO_2_, silica-based, TiO_2_, and Ag NPs) because they play a pivotal role in dental applications, serving as dental fillings, enamel surface polish to deter caries, and implant materials surpassing conventional alternatives in efficacy [16]. NPs have been used for medical and environmental applications. Nanotechnology has revolutionized the healthcare sector, enhancing diagnosis accuracy, monitoring diseases, advancing surgical equipment, promoting regenerative medicine, refining vaccine development, and optimizing medication delivery systems. Moreover, it facilitates the development of cutting-edge research instruments, paving the path for the creation of innovative drugs to enhance treatments across a spectrum of ailments [17]. Also, nanotechnology holds immense promise in offering inventive solutions to diverse environmental challenges. These encompass enhanced pollution reduction techniques, advanced water treatment methods, precise environmental sensing technologies, efficient remediation processes, and the optimization of alternative energy sources to be more economically viable. Engineered nanomaterials possess distinctive properties that facilitate the development of these innovative technologies, paving the way for sustainable solutions to environmental issues [18].

The dentistry use of nanotechnology not only garners patient interest for its potential cost-effectiveness and time-saving attributes but also offers psychological relief by minimizing treatment-related stress. The ongoing advancement of tailored nanomaterials holds promise in resolving dental issues. Although nanotechnology’s current impact on oral disease treatment is somewhat constrained, ongoing research endeavors are poised to unlock significant advancements in the near future [16,19]. NPs are materials with dimensions from 1 to 100 nm [20], and certain NPs exhibit antimicrobial properties, curtailing bacterial proliferation [16]. The synthesis of Cu-based nanoparticles can be achieved through chemical or biological methods. The chemical method typically follows the “bottom-up” and “top-down” approaches. In the bottom-up method, atomic-level precursors are utilized to synthesize nanoscale materials. The top-down approach involves breaking down a bulk solid into progressively smaller components to obtain nanoparticles [21]. Furthermore, green synthesis routes have been utilized for both the enzymatic and non-enzymatic production of Cu NPs. These methods involve the interaction of copper salt with organic compounds, resulting in Cu NP formation. This green method has several advantages, including easy accessibility, non-toxicity, cost-effectiveness, and straightforward handling [22]. 

Similar to other metal nanoparticles employed in dentistry [16], Cu NPs have diverse sizes and forms, alongside a distinctive distribution and an impressive surface-area-to-volume ratio [19,23]. These characteristics enhance the bio-physiochemical functionalization, antimicrobial efficacy, and biocompatibility of these nanoparticles [19]. Studies have demonstrated that copper oxide (CuO) nanoparticles have notable antimicrobial properties and effectively impede biofilm formation [24]. Furthermore, Cu presents advantages over other metals because of its abundant and relatively inexpensiveness, making Cu NPs inexpensive for large-scale applications [19,25]. 

The toxicity mechanism of Cu NPs involves their interaction, accumulation, and subsequent dissolution within the cellular membrane, leading to alterations in membrane permeability. Also, the release of ions from NPs induces the generation of reactive oxygen species (ROS), triggering lipid peroxidation, protein oxidation, and DNA degradation [26,27]. Furthermore, the metal ions present within the cell inflict damage upon DNA and interfere with ATP production. Cu ions, specifically, exhibit interactions with the phosphate and thiol groups present in proteins and DNA, resulting in denaturation and other structural disturbances [23]. Notably, Cu NPs have exhibited superior bactericidal activity against *E. coli*, *B. subtilis*, and *S. aureus* compared with silver NPs, which are commonly employed in biomedical research [14,15]. In oral applications, it has been reported that Cu NPs can be added to dental cement, restorative materials, adhesives, resins, irrigating solutions, obturations, orthodontic archwires and brackets, implant surface coatings, and the bone regeneration process [15,16,17,18,19,20]. Despite these uses in dentistry, the effect of Cu NPs has been principally studied regarding *S. mutans* [28,29,30], and little is known about their effect on commensal bacteria. 

Furthermore, studies should explore strategies to maximize the therapeutic efficacy of Cu NPs against pathogenic bacteria while minimizing their adverse effects on commensal bacteria. This may involve optimizing NP sizes, surface chemistry, and dosages to selectively target pathogens while preserving commensal populations.

In this study, we explored the effects of three distinct Cu NPs on relevant cavity-associated bacteria (*S. mutans* and *L. rhamnosus*), as well as healthy oral-associated bacteria (*S. sanguinis* and *S. salivarius*) in vitro. We assessed the influence of Cu NPs on the growth, viability, and biofilm formation of both commensal and pathobiont oral bacteria. Our findings revealed that CuO exhibited a pronounced inhibitory effect against the pathogenic bacteria tested. These results suggest the potential for developing strategies to enhance the therapeutic efficacy of Cu NPs against pathogenic bacteria while mitigating their adverse effects on commensal bacteria.

## 2. Methods 

### 2.1. Nanoparticle Characterization

Metallic copper nanoparticles (Cu^0^, Cu_2_O, and CuO) were obtained from NANOTEC S.A. (Santiago, Chile) [31,32] Nanoparticles with sizes ranging from ~40 to 70 nm were used in all the experiments (99.9978% purity). These NPs were made using chemical methods The characteristics of the Cu NPs used in this work are detailed in Table 1.

### 2.2. Bacterial Cultures

The *Streptococcus* genus bacteria (*S. mutans* ATCC 25175; *S. salivarius* ATCC 13419; and *S*. *sanguinis* SK36) were grown in Brain Heart Infusion broth (BHI)–bacitracin (0.2 units/mL). The solid medium utilized was based on tryptone, yeast extract, cystine (TYC) agar, bacitracin, and sucrose (5%). *L. rhamnosus* ATCC 53103 was grown in a liquid De Man, Rogosa, and Sharpe (MRS) medium. The solid growth was on MRS-Agar. All bacteria were grown at 37 °C in microaerophilic conditions (candle jars).

### 2.3. Antibacterial Activity of Nanoparticles

*S. mutans*, *L. rhamnosus*, *S. salivarius*, and *S. sanguinis* minimal inhibitory concentrations (MIC) for each Cu NP were determined in a planktonic state. Briefly, 1 × 10^5^ cells/mL initial inoculums were used, and bacteria were grown at increasing concentrations of Cu NPs (200–1000 µg/mL) and in the absence of NPs as a positive control. Samples were incubated for 48 h at 37 °C with constant shaking (90 rpm), and OD_600_ was measured. All experiments were conducted in triplicate to ensure the robustness and reliability of the results.

### 2.4. Viability Assays

The effect of CuO NPs on the viability of *S. mutans*, *S. sanguinis*, and *L. rhamnosus* viability was determined by utilizing 1 × 10^5^ cells/mL as initial inoculums in BHI (*Streptococcus*) and MRS (*Lactobacillus*) media amended with 100, 300, or 500 µg/mL of CuO NPs. Growth in the absence of NPs was used as a positive control. After 48 h of growth at 37 °C with constant shaking (90 rpm), bacteria were seeded in solid medium for a CFU count. 

### 2.5. Anti-Biofilm Activity of Nanoparticles

Biofilm formation for each bacterial strain was assayed in a BHI medium supplemented with sucrose (10%) in 96-well plates following a previously described protocol [33]. In total, 1 × 10^5^ cells/mL initial inoculums were used. Samples were incubated for 48 h at 37 °C with shaking (90 rpm), and OD_600_ was measured. Agitation facilitates the better dispersion of nanoparticles within a solution, ensuring more uniform exposure of the biofilm to NPs [34]. Planktonic cells were separated from sessile cells, and the OD_600_ of the supernatants was analyzed as in [35]. Finally, we analyzed the Biofilm Formation Index (BFI = OD_570_/OD_600_) [35]. The biofilm experiment was conducted with a minimum of three independent replicates to validate the consistency and reproducibility of the results.

### 2.6. Anti-Biofilm Activity of Nanoparticles on Tooth Crowns

The use of extracted human molars and third molars was approved by the Ethics Committee at University Andres Bello (approval number: 001/2019). Immediately after extraction, the teeth were thoroughly cleaned using curettes, and the crowns of the teeth were separated from the roots. Then, tooth crowns were immersed for 10 min in a 4.9% chlorine solution, washed with sterile DI water, and autoclaved. Then, 24-well plates containing 70% pasteurized saliva [36], 30% BHI, and 10% sucrose [37] were used for bacterial growth and biofilm formation. Teeth were pre-incubated for 4 h in this solution [37]. Then, *S. mutans* were inoculated (1 × 10^5^ cells/mL) and incubated for 48 h in the presence of NPs (200 µg/mL) at 37 °C and constant agitation (90 rpm). A negative control for biofilm formation was used in the presence of NPs and without *S. mutants* added.

For the biofilm disruption assay, *S. mutans* was inoculated (1 × 10^5^ cells/mL) and incubated for 48 h at 37 °C, without agitation. Consequently, when the mature biofilm was formed, Cu NPs were added, and the cells were incubated for 24 h at 37 °C with constant shaking (90 rpm). At the end of each experiment, each tooth crown was washed with sterile deionized water and analyzed via scanning electron microscopy (SEM). The biofilm experiment on teeth was conducted in two independent replicates due to the challenge of obtaining healthy teeth amidst the global COVID-19 pandemic conditions.

### 2.7. Scanning Electron Microscopy (SEM) Visualization

Twelve dental pieces were fixated in 2.5% Glutaraldehyde with 0.1 M Sodium Cacodylate Buffer for 2 h. The samples were washed 3 times for 5 min in distilled water and prepared as in [29], and they were finally visualized using a scanning electron microscope (Jeol Model JSM IT300LV, Tokyo, Japan).

### 2.8. Statistical Analysis

Statistical analysis was performed using GraphPad 7.0a. A two-way analysis of variance (ANOVA) with Dunnett’s multiple comparison test was used, and significant results were considered with a *p*-value < 0.05.

## 3. Results

### 3.1. Effect of Cu NPs on the Growth of Pathogenic and Commensal Oral Bacteria

To determine the effect of Cu NPs on the growth of pathogenic and commensal oral bacteria, growth assays and MIC determinations were performed on *S. mutans*, *L. rhamnosus*, *S. salivarius*, and *S. sanguinis* in the presence of three types of Cu NPs. In pathobiont bacteria, *S. mutans* exhibited a significant decrease in growth in the presence of Cu^0^ NPs. Specifically, a ~46% decrease in growth was observed at a concentration of 400 µg/mL of Cu^0^ NPs; however, the minimum bactericidal concentration (MBC) was not determined, even after testing up to 2000 µg/mL. Similarly, exposure to 400 µg/mL of Cu_2_O NPs resulted in a ~48% decrease in growth, with no growth observed at the MBC of 500 µg/mL Cu_2_O NPs. Additionally, exposure to 200 µg/mL of CuO NPs led to a ~56% decrease in growth, with the MBC determined to be 400 µg/mL of CuO NPs.

In *L. rhamnosus*, a ~46% reduction in growth was observed at a concentration of 200 µg/mL of Cu^0^ NPs, with the minimal bactericidal concentration (MBC) determined to be 800 µg/mL of NPs (Figure 1B, black bar). For Cu_2_O NPs (Figure 1B, light-gray bars), a ~70% decrease in bacterial count was noted at 200 µg/mL, with the MBC observed at 400 µg/mL. Additionally, with CuO NPs, a ~29% reduction in bacterial count was detected at 100 µg/mL, with the MBC determined to be 200 µg/mL.

For the commensal bacterium *S. sanguinis*, no significant differences in growth were detected in the presence of Cu^0^ NPs (Figure 1C, black bar) until a concentration of 2000 µg/mL was reached. With Cu_2_O NPs, there was a ~58% reduction in growth observed at 800 µg/mL (Figure 1C, light-gray bar), with a complete absence of growth observed at 1000 µg/mL. Lastly, CuO NPs resulted in a ~63% decrease in growth at 600 µg/mL (Figure 1C, dark-gray bar), with no growth observed at 800 µg/mL. Interestingly, *S. sanguinis* exhibited resistance twice and four-fold higher to Cu_2_O and CuO compared with *S. mutans* and *L. rhamnosus*, respectively.

In *S. salivarius*, identical to *S. sanguinis*, no differences were observed in the presence of Cu^0^ NPs (Figure 1D, black bars), even at 2000 µg/mL. With Cu_2_O NPs, the growth diminished by ~32% at 200 µg/mL, and no growth was observed at 400 µg/mL (Figure 1D, light-gray bars). Furthermore, with CuO NPs, the growth decreased by ~43% at 400 µg/mL, with no growth observed at 600 µg/mL (Figure 1D, dark-gray bars).

Based on these results, the MICs for each type of Cu NP were determined (Table 1). Our findings indicate that all Cu NPs affected each bacterial species differentially. Generally, copper oxide (Cu_x_O) NPs demonstrated a greater impact on cell density for three of the tested bacteria (excepting *S. sanguinis*), while Cu^0^ NPs exhibited the lowest effect for all bacteria, with *L. rhamnosus* being the most sensitive. Interestingly, CuO NPs showed higher toxicity toward pathobionts compared with commensal bacteria (Table 2).

### 3.2. Effect of CuO NPs on the Viability of Caries-Associated Bacteria

Since the growth assays revealed that *S. sanguinis* is the most tolerant bacteria and, interestingly, CuO NPs had a greater effect on pathogenic bacteria, we proceeded to analyze their effects on bacterial viability. We compared the response of the highly tolerant *S. sanguinis* to that of pathogenic bacteria using 100, 300, and 500 µg/mL concentrations of CuO NPs.

After exposing *S. sanguinis* to 100 µg/mL of CuO nanoparticles (NPs), we detected 7 × 10^7^ CFU/mL, representing 47% of the cells that remained alive relative to the control without CuO NPs. At a concentration of 300 µg/mL, cell growth decreased to 2 × 10^7^ CFU/mL (11%) after exposure and further decreased to 8 × 10^6^ CFU/mL (6%) after exposure to 500 µg/mL (Figure 2, dark-gray bars).

*S. mutans* exhibited a viability of 5.5 × 10^8^ CFU/mL (22%) after exposure to 100 µg/mL of CuO NPs and decreased to 9 × 10^6^ CFU/mL (0.2%) at 300 µg/mL, and no viable bacteria were observed at 500 µg/mL (0%) (Figure 2, black bars).

Finally, in *L. rhamnosus*, we detected 1.5 × 10^8^ CFU/mL (12.5%) after exposure to 100 µg/mL of CuO NPs, which decreased to 1.6 × 10^5^ CFU/mL (0.01%) at 300 µg/mL and further decreased to 6 × 10^3^ CFU/mL at 500 µg/mL, illustrating a near 100% decrease in viability (Figure 2, light-gray bars).

In summary, these results indicate that CuO NPs strongly affect the viability of *S. mutans* and *L. rhamnosus*, while exhibiting a minor effect on the survival rate of *S. sanguinis.*

### 3.3. Anti-Biofilm Activity of Cu NPs over Oral Bacteria

Dental cavities develop from a polymicrobial biofilm that forms on solid surfaces, such as enamel. Therefore, it is crucial to examine whether the ability of oral bacteria to attach to surfaces is altered in the presence of Cu NPs. To assess this, the Biofilm Factor Index (BFI = OD_570_/OD_600_) was determined for the *Streptococcus* genus (Figure 3). We omitted the analysis for *L. rhamnosus* as it was unable to form a biofilm under these conditions resulting in noncomparable results.

Significant decreases in adherence of approximately 95%, 96%, and 94% were observed for *S. mutans* in the biofilm assay when grown in BHI–sucrose at 100, 200, and 300 µg/mL concentrations of Cu^0^ NPs, respectively (Figure 3A, black bars). In Cu_2_O NPs, reductions in biofilm formation of approximately 58%, 81%, and 79% were noted at concentrations of 100, 200, and 300 µg/mL, respectively (Figure 3A, light-gray bars). Finally, following exposure to 100, 200, and 300 µg/mL of CuO NPs, the BFIs of *S. mutans* decreased by approximately 50%, 80.5%, and 82%, respectively (Figure 3A, dark-gray bars). 

Surprisingly, we did not observe any clear negative effect on adherence in commensal bacteria. In the presence of Cu^0^ NPs (Figure 3B, black bars), *S. sanguinis* showed no significant changes at any concentration tested, except for the highest concentration (300 µg/mL) of Cu_2_O and CuO NPs, which decreased the BFI by approximately 43% and 48%, respectively. 

Interestingly, *S. salivarius* exhibited increased biofilm formation with Cu NPs. At 200 and 300 µg/mL concentrations of Cu^0^ NPs, the BFI increased by approximately 221% and 378%, respectively. Similarly, with 200 and 300 µg/mL of Cu_2_O NPs, increases of approximately 226% and 300% were observed, respectively. Lastly, at 200 and 300 µg/mL concentrations of CuO NPs, there were increases of approximately 232% and 247%, respectively. 

In general, this biofilm assay conducted on the *Streptococcus* genus in a comparative manner indicated that oxide Cu NPs can decrease the biofilm formation of the oral commensal bacterium *S. sanguinis*, albeit only at higher concentrations, and surprisingly, an increase in adherence was detected in *S. salivarius*. Finally, even in the presence of high concentrations of sucrose, a strong inhibitory effect on biofilm formation was observed in *S. mutans*, demonstrating its high sensitivity to the presence of Cu NPs.

### 3.4. Anti-Biofilm Effect of Cu NPs against S. mutans on Tooth Crowns

To assess whether the observed anti-biofilm effect (Figure 3) also occurs under physiological conditions, biofilm formation on extracted human teeth was evaluated using scanning electron microscopy (SEM), focusing on the principal cariogenic bacterium, *S. mutans* (Figure 4). This assay utilized healthy extracted human teeth and pasteurized saliva to simulate real-life conditions for potential application. Due to the in vitro results obtained from the 96-well plate assay indicating no significant effect on *S. salivarius* and *S. sanguinis* (Figure 3) and the challenges associated with obtaining healthy teeth during the global COVID-19 pandemic, the analysis was not conducted for commensal bacteria.

In the absence of Cu NPs, we observed aggregated cocci over an extracellular polymeric substance (EPS) structure on all teeth (Figure 4A), indicating the formation of a regular biofilm. However, in the presence of Cu^0^ NPs, honeycomb-like structures representing EPS indicating natural cellular detachment from the biofilm at this time [38] were observed (Figure 4B, black arrow), with fewer bacterial aggregates (Figure 4B, white arrow). With oxide Cu NPs (Cu_x_O NPs), a reduced biofilm was observed (Figure 4C,D), characterized by only a few amorphous bacteria present on the tooth surface (white arrow, Figure 4C,D). Consequently, Cu^0^ NPs decreased biofilm formation, while Cu_x_O NPs showed an unclear biofilm structure formation at the concentrations tested compared with the control in Figure 4A.

Additionally, we evaluated the effect of NPs on biofilms already formed on tooth crowns (Figure 5), as described in the materials and methods section. After 48 h of incubation, *S. mutans* formed a mature biofilm on the tooth surface (Figure 5A, white arrow). By 72 h, the presence of EPS (Figure 5B, black arrow) and aggregated bacteria (Figure 5B, white arrow) was easily detected, along with a few small honeycomb-like structures (Figure 5B, black arrow). Upon the addition of Cu^0^ NPs (Figure 5C), fewer aggregated bacteria were observed (Figure 5C, white arrow), along with reduced EPS and honeycomb-like structures (Figure 5C, white arrow). After 72 h of mature biofilm formation, the addition of Cu_2_O NPs (Figure 5D) showed few bacterial cells (white arrow), alongside increased EPS and honeycomb-like structures (Figure 5D, black arrow). Notably, these structures were not observed in the “biofilm formation assay” previously shown with Cu_2_O NPs (Figure 4C), suggesting that these NPs may disrupt the biofilm and induce the detachment of cells without affecting biofilm formation at this concentration. Finally, the detachment effect was more pronounced in biofilms exposed to CuO NPs, where practically no EPS structures or bacteria were observed (Figure 5E), indicating that the exposure of biofilms formed on tooth crowns to CuO NPs led to the complete disruption of the biofilms. Overall, our results indicate that Cu_x_O NPs strongly decrease biofilm formation and promote biofilm detachment from tooth crowns, with the total absence of biofilm and related structures observed upon CuO NP exposure.

This study presents the first comprehensive investigation of the effect of various Cu NPs on the growth and biofilm formation of four significant oral bacteria, revealing the distinctive impact of Cu NPs on each bacterial strain. CuO NPs emerged as the most potent in inhibiting both the growth and biofilm formation of pathogenic oral bacteria. While further competition assays involving all strains and viability/biofilm formation assays using oral samples such as saliva are essential, our findings provide a valuable foundation for the potential utilization of CuO NPs as anti-cavity agents, particularly for their pronounced effect on pathogenic bacteria.

## 4. Discussion

Worldwide, untreated cavities represent the most prevalent oral health concern. This disease poses a significant global health challenge, emphasizing the urgent need for its control [1]. Copper (Cu) and copper oxide nanoparticles (Cu_x_O NPs) have emerged as focal points of research in biomedical applications, drawing attention because of their multifaceted advantages, which include enhancing drug stability, facilitating precise biodistribution, elevating therapeutic efficacy, and facilitating targeted delivery to specific sites through active or passive targeting mechanisms [39]. Added to medical applications, their remarkable antimicrobial properties make them a cornerstone of various industries worldwide. Harnessing the potential of Cu-based nanomaterials extends their applications across diverse sectors, such as agriculture, livestock management, water treatment, wood preservation, and textile manufacturing [40]. Moreover, the remarkable conductivity and cost-effectiveness of Cu nanomaterials position them as compelling alternatives to noble metal counterparts in pivotal fields like solar energy conversion, battery technology, and electrochemical sensing [40,41].

In this way, Cu NPs have been utilized to inhibit the growth of oral pathobionts in various dental materials, including metals and alloys; polymers and resins; cements; and other miscellaneous materials [15]. Because cavities result from oral dysbiosis characterized by an increase in acidogenic and aciduric bacteria [42], AOAs used to control the growth of these species should also promote the maintenance of a balanced microbial oral community, or eubiosis [43]. Currently, several commonly used antimicrobial agents against pathogenic oral bacteria include sodium fluoride, chlorhexidine, penicillin, chitosan, and daptomycin. Interestingly, *S. sanguinis* exhibits greater sensitivity to most standard antimicrobial oral agents compared with *S. mutans*, contrasting with the situation observed with Cu NPs (Appendix A). Only chlorhexidine (CHX) and penicillin appear to be more toxic for *S. mutans*. However, it has been documented that CHX-containing mouthwash can alter the salivary microbiome, leading to a more acidic environment and reducing nitrite availability in healthy individuals [12]. Regarding antibiotics such as penicillin, studies have reported alterations in the oral microbiota, which may subsequently affect the concentration of salivary antibodies [13]. 

With the aim of evaluating the effects of Cu NPs on some important representative bacteria from the oral microbiome, we conducted in vitro analyses to assess the impact of three types of Cu NPs on *S. mutans*, *L. rhamnosus*, *S. sanguinis*, and *S. salivarius.*

In planktonic lifestyle, we observed that Cu_x_O NPs exhibited a stronger antibacterial effect compared with Cu^0^, with MIC values ranging between 400 and 600 µg/mL in *S. mutans* and *S. salivarius*, while surprisingly showing high tolerance in *S. sanguinis* (MIC > 800 µg/mL). Interestingly, CuO NPs exerted a pronounced effect on cariogenic bacteria compared with commensal ones. This observation aligns with the viability assay, where *S. sanguinis* exhibited higher survival rates (11.4%) compared with the minimal cell survival observed in *S. mutans* and *L. rhamnosus* (less than 0.2%).

Previous studies have reported that CuO NPs induce higher levels of reactive oxygen species (ROS) than Cu_2_O and Cu^0^ NPs. CuO NPs generate ROS through Haber–Weiss and Fenton-type reactions, while Cu_2_O NPs only generate ROS through Fenton-type reactions [44]. Moreover, it has been reported that CuO NPs demonstrate a higher degree of internalization and better activity at lower concentrations [25]. This phenomenon could explain the antibacterial effect observed with CuO NPs. Additionally, considering that CuO NPs do not exhibit toxic effects on human cells even at high concentrations (up to 5000 µg/mL) [45], these NPs could be suitable candidates for AOA applications.

In Cu^0^ NPs, all Streptococcus genus bacteria exhibited high tolerance (MIC > 1000 µg/mL), with *L. rhamnosus* showing slight sensitivity (MIC 800 µg/mL). Previous studies have reported the greater antibacterial activity of Cu^0^ NPs compared with other Cu NPs in aerobic conditions, attributable to their stronger ability to accept electrons, leading to bacterial membrane rupture [46], and their superior capacity to release Cu^+^ ions, facilitating contact killing activity against bacteria [47]. However, our studies were conducted under microaerophilic conditions, where there is likely less oxidation, potentially reducing the release of Cu^+^ ions and altering the lethal contact effect.

The most tolerant strain to all Cu NPs was *S. sanguinis*. Although there are no reports of this bacterium being exposed to Cu NPs, it has shown resistance to high concentrations of copper salts (MIC, 1000 µg/mL) [48]. Previously, a decrease in Cu^+^ released in the presence of H_2_O_2_ has been reported [49]. The decrease in Cu released by NPs may occur because the OH radicals generated are adsorbed on the nanocrystal surface [49]. In this way, the production of H_2_O_2_ by *S. sanguinis* [9] could potentially protect this bacterium from the effects of Cu NPs.

In contrast, Cu NPs strongly affected *L. rhamnosus*. The sensitivity of *L. rhamnosus* to Cu may be attributable to amine and carboxyl groups on the cell surface, which have a greater affinity for the metal [50]. Generally, *Lactobacillus* strains display an electronegative charge with a cell surface dominated by anionic compounds [51], which could enhance the binding of Cu NPs (and Cu ions), leading to damage to the cell membrane.

In the biofilm assay, we did not detect a significant effect of Cu NPs on biofilm formation in *S. sanguinis*, and surprisingly, an increase in biofilm formation in *S. salivarius* was observed. To date, no studies reporting a positive effect of Cu NPs on biofilm formation have been published. However, the exposure of *E. gracilis* to Cu^2+^ stimulated biofilm formation, suggesting that biofilm formation could be considered a protective mechanism [52]. Cu NPs significantly reduced biofilm formation in *S. mutans* in all assays. On tooth crowns, we observed a decrease in biofilm structures in the presence of all NPs, consistent with previous reports involving Cu-containing NPs [29]. Honeycomb-like structures were observed in biofilms exposed to Cu^0^ NPs, indicating cell detachment from the biofilm [38]. Thus, it can be speculated that Cu^0^ NPs did not block attachment to the tooth but instead caused bacterial release [38].

In the case of CuO NPs, no EPS structures were observed, indicating that the oxides affected the formation of glucan matrix even in the presence of sucrose. A previous study showed that copper ions suppressed the expression of certain glucosyltransferase, *gtf* genes, which code to the enzymes responsible for synthesizing glucans from sucrose. These genes are crucial for the formation of glucan matrix in dental plaque, contributing to biofilm formation and dental caries development, thereby negatively affecting biofilm formation in *S. mutans* [53]. Additionally, it has been reported that biofilm formation is regulated by autoinducers (AIs) mediated by quorum sensing (QS) [54]. In 2021, it was demonstrated that copper inhibited the QS of *S. agalactiae* [55]. In *S. mutans*, Cu NPs could negatively affect biofilm formation through both of these mechanisms.

## 5. Conclusions

The impact of Cu NPs on key species within representative oral microbiome bacteria was thoroughly investigated, with a focus on fostering oral health (eubiosis). Our research uncovered the distinct effect of CuO NPs, showing the significant inhibition of pathogenic bacterial growth while demonstrating minimal influence on representative beneficial commensal species. Particularly noteworthy was the ability of CuO NPs to effectively deter biofilm formation, with limited adverse effects on the commensal bacteria tested. This was evidenced in biofilm analyses conducted on tooth surfaces, where both the prevention of biofilm formation and the disruption of already established mature biofilms were observed.

Moving forward, our research aims to expand into in vivo studies to analyze the in vivo effect of these Cu NPs. Overall, our findings underscore the potential of CuO NPs for targeted applications against oral pathogens, emphasizing their promise as a therapeutic strategy for enhancing oral health.

## Figures and Tables

**Figure 1 microorganisms-12-00624-f001:**
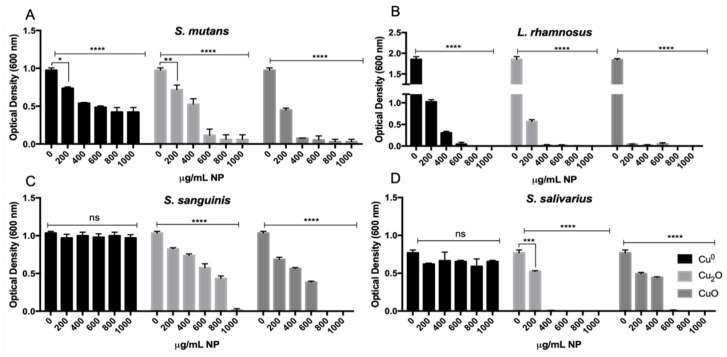
Oral bacteria growth analysis (OD_600_) in the presence of different concentrations of Cu NPs. (**A**) *S. mutans*, (**B**) *L. rhamnosus*, (**C**) *S. sanguinis*, and (**D**) *S. salivarius*. The experiment was developed in an adequate medium for each strain. Bacteria were grown for 48 h at 37 °C and 90 rpm in the presence of Cu^0^, Cu_2_O, or CuO NPs. We used the absence of NPs as a positive control of growth. Data represent the mean ± SEM of DO_600_ values obtained in three independent experiments performed in triplicate. Asterisks represent statistically significant values (ns: no significative; * *p* < 0.05; ** *p* < 0.01; *** *p* < 0.001; **** *p* < 0.0001) compared with the related controls without NPs.

**Figure 2 microorganisms-12-00624-f002:**
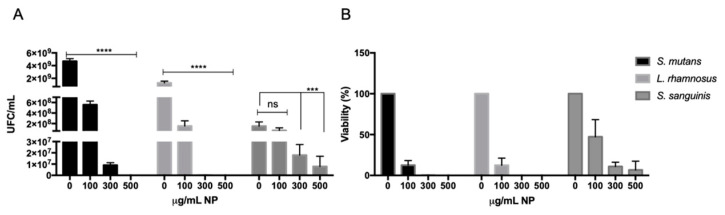
Assessment of viable cells after exposure to 100, 300, and 500 µg/mL of CuO NPs. A viability assay was developed in an adequate liquid medium for 48 h at 37 °C and 90 rpm in the presence of CuO NPs. Then, bacteria were seeded in adequate agar plaque for (**A**) Colony-Forming Unit (CFU) determination. (**B**) Viability percentage with respect to growth in the absence of CuO NPs in each strain. We used the absence of NPs as a positive control of growth. Data represent the mean ± SEM of CFU/mL values obtained in three independent experiments performed in duplicate. Asterisks represent statistically significant values (ns: no significative; *** *p* < 0.001; **** *p* < 0.0001) compared with the controls.

**Figure 3 microorganisms-12-00624-f003:**
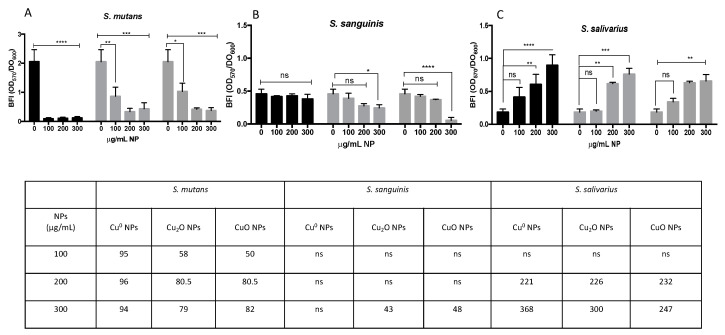
In vitro biofilm formation assay. (**A**) *S. mutans*, (**B**) *S. sanguinis*, and (**C**) *S. salivarius* were grown independently in the presence of different Cu NPs. Biofilm formation was analyzed with a crystal violet assay after 48 h of growth, and variation in biofilm formation was determined through the Biofilm Formation Index (DO_570_/DO_600_). Data represent the mean ± SEM of BFI values obtained in three independent experiments performed in triplicate. Asterisks represent statistically significant values (ns: no significative; * *p* < 0.05; ** *p* < 0.01; *** *p* < 0.001; **** *p* < 0.0001) compared with the related controls. The table at the bottom shows the percentage increase or decrease in biofilm formation.

**Figure 4 microorganisms-12-00624-f004:**
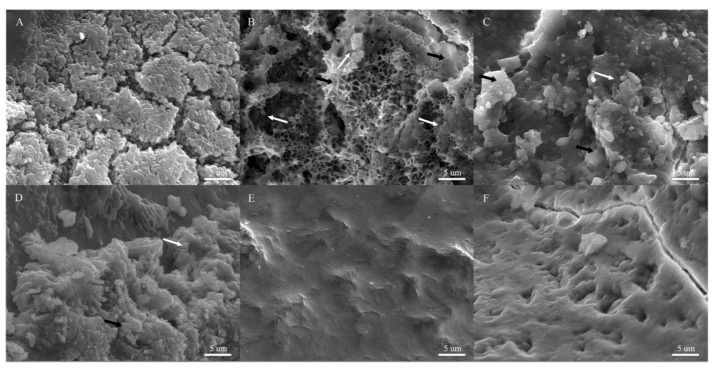
Effect of Cu NPs on *S. mutans* biofilm formation on tooth crown. Scanning electron micrography images of *S. mutans* biofilms on the crown of tooth surface for (**A**) bacteria grown on a tooth immersed in medium without Cu NPs; (**B**) bacteria grown on a tooth immersed in medium exposed to 200 µg/mL of Cu^0^ NPs, (**C**) 200 µg/mL of Cu_2_O NPs, and (**D**) 200 µg/mL of CuO NPs; and (**E**,**F**) controls without bacteria. White arrow: *S. mutans*; black arrow: EPS.

**Figure 5 microorganisms-12-00624-f005:**
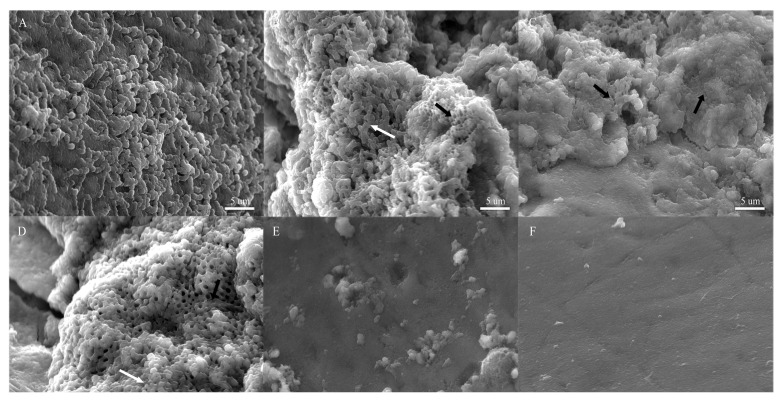
Effect of Cu NPs on *S. mutans* biofilms formed on tooth crown. Scanning electron micrography images of tooth crowns after (**A**) 48 h of biofilm formation without Cu NPs (mature biofilm); (**B**) 72 h of biofilm formation without Cu NPs (positive control); and mature biofilm exposed 24 h to (**C**) 200 µg/mL of Cu^0^ NPs, (**D**) 200 µg/mL of Cu_2_O NPs, and (**E**) 200 µg/mL of CuO NP. (**F**) Tooth crown without bacteria: negative control. White arrow: *S. mutans*; black arrow: EPS.

**Table 1 microorganisms-12-00624-t001:** Characteristics of the Cu NPs used in this work.

	Cu^0^ NPs	Cu_2_O NPs	CuO NPs
CAS number	7440-50-8	1317-38-0	1317-38-0
Molecular weight	63.5 g/mol	143.9 g/mol	79.6 g/mol
Color	Brown-red	green	black
Size	~40–70 nm	~40–60 nm	~40–60 nm
Batch	180314-RN	230315-SP	190726-MA

**Table 2 microorganisms-12-00624-t002:** Minimal inhibitory concentration (MIC) values (µg/mL) of Cu^0^, Cu_2_O, and CuO NPs to *S. mutans*, *L. rhamnosus*, *S. sanguinis*, and *S. salivarius*.

	MIC Cu^0^ NPs (μg/mL)	MIC Cu_2_O NPs (μg/mL)	MIC CuO NPs (μg/mL)
*S. mutans*	>1000	500	400
*L. rhamnosus*	800	300	200
*S. sanguinis*	>1000	1000	800
*S. salivarius*	>1000	400	600

## Data Availability

The original contributions presented in this study are included in the article and the associated Appendix A.

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
