# Peer review of "Antimicrobial Effect of Copper Nanoparticles on Relevant Supragingival Oral Bacteria"

_microorganisms, 2024, doi:10.3390/microorganisms12030624_

Round 1

Reviewer 1 Report

Comments and Suggestions for Authors

Dear Authors,

the study is interesting but some missing points you have.

1. Please add in Introduction information about nanoparticles in medicine and environment and contemporary opinions about that. It should be also discussed in Discussion chapter.

2. Are there any other metal nanoparticles used in medicine/dentistry?

3. Are there any products / materials already in dentistry with Cu?

4. The aim is already the conclusion, it should be redacted.

Author Response

Dear Authors, the study is interesting but some missing points you have.

  1. Please add in Introduction information about nanoparticles in medicine and environment and contemporary opinions about that. It should be also discussed in Discussion chapter.

Response: Done. As requested, the ms was modified (see new lines: 86-97).

“NPs has been used in medical and environmental applications. Nanotechnology revolutionizes the healthcare sector enhancing diagnosis accuracy, monitoring diseases, advancing surgical equipment, promoting regenerative medicine, refining vaccine development, and optimizing medication delivery systems. Moreover, it facilitates the development of cutting-edge research instruments, paving the path for the creation of innovative drugs to enhance treatments across a spectrum of ailments (Haleem et al. 2023). Also, nanotechnology holds immense promise in offering inventive solutions to diverse environmental challenges. These encompass enhanced pollution reduction techniques, advanced water treatment methods, precise environmental sensing technologies, efficient remediation processes, and the optimization of alternative energy sources to be more economically viable. Engineered nanomaterials possess distinctive properties that facilitate the development of these innovative technologies, paving the way for sustainable solutions to environmental issues (Pathakoti, Manubolu, and Hwang 2018).”

  1. Are there any other metal nanoparticles used in medicine/dentistry?

Response: Done. The information requested was incorporated in the ms (see new lines 83-84).

“In this context, different types of nanoparticles (NPs) have been applied in dental materials (Yun et al. 2022)(FeOx, ZrO2, Silica-based, TiO2 and Ag NPs) because play a pivotal role in dental applications, serving in dental fillings, enamel surface polishing to deter caries, and as implant materials surpassing conventional alternatives in efficacy (Priyadarsini, Mukherjee, and Mishra 2018).”

  1. Are there any products / materials already in dentistry with Cu?

Response: Done. The information requested was incorporated in the ms (see new lines 128-130).

“In oral applications, it was reported that Cu NPs can be added to dental cements, restorative materials, adhesives, resins, irrigating solutions, obturations, orthodontic archwires and brackets, implant surface coatings, and the bone regeneration process (Priyadarsini et al. 2018).”

  1. The aim is already the conclusion; it should be redacted.

Response: Done. The ms was modified properly (see new lines 446-458).

The authors are requested to revise the manuscript as per the suggestions provided above before further processing for consideration in this Journal.

Response: Done.

Comments on the Quality of English Language

Moderate editing of English language required

Response: Done.

Reviewer 2 Report

Comments and Suggestions for Authors

The following comments can aid authors to improve the quality of manuscript:

·         In abstract, change antimicrobial to antibacterial.

·         In introduction, define the acronym of NPs. In addition, give a brief overview of nanoparticles and their advantages as AOAs.

·         In introduction, explain the advantages of copper over other metal ions, synthesis methods, and what has been documented in the synthesis of nanomaterials for antibacterial applications.

·         Please add the ATCC number of the cultured strains in this study.

·         Explain what the positive control was used in this study and included in Figure 1 and 2.

·         The explanation about the effect of the antibacterial activity of CuNPs can be associated with the physical and chemical data of these nanomaterials. It would be important to consult this with suppliers.

·         Based on the scope of this work, it would be beneficial to include the effect of CuNPs in the viability of healthy cell lines.

·         The list of abbreviations is incomplete, please revise for quorum sensing, autoinducers,  minimal inhibitory concentration, minimal bactericidal concentration, among others.

Author Response

Comments and Suggestions for Authors

The following comments can aid authors to improve the quality of manuscript:

In abstract, change antimicrobial to antibacterial.

Response: Done. See new line 31.

In introduction, define the acronym of NPs. In addition, give a brief overview of nanoparticles and their advantages as AOAs.

Response: Done. See new line 83 (NPs) and lines 98-102(Advantages as AOAs).

In introduction, explain the advantages of copper over other metal ions, synthesis methods, and what has been documented in the synthesis of nanomaterials for antibacterial applications.

Response: Done. See new lines 118-119 (advantages) and lines 104-111 (methods) of the ms.

Please add the ATCC number of the cultured strains in this study.

Response: Done. Included in lines 150, 151 and 153.

Explain what the positive control was used in this study and included in Figure 1 and 2.

Response: Done. The positive control of growth in absence of NPs was included in the Figures and the corresponding legends (lines 218 and 261). In addition, we included more details in materials and methods section (lines 160 and 167).

The explanation about the effect of the antibacterial activity of CuNPs can be associated with the physical and chemical data of these nanomaterials. It would be important to consult this with suppliers.

Response. We agree with the reviewer on this point. Accordingly, we discussed this possibility in the new version of the ms (see new introduction lines 120-126 and discussion lines 400, 411, 416).

“The toxicity mechanism of Cu NPs involving their interaction, accumulation, and subsequent dissolution within the cellular membrane, leading to alterations in membrane permeability. Also, the release of ions from the NPs induces the generation of reactive oxygen species (ROS), triggering lipid peroxidation, protein oxidation, and DNA degradation (Chatterjee, Chakraborty, and Basu 2014; Raffi et al. 2010). Furthermore, the metal ions present within the cell inflict damage upon DNA and interfere with ATP production. Cu ions, specifically, exhibit interactions with phosphate and thiol groups present in proteins and DNA, resulting in denaturation and other structural disturbances (Ramos-Zúñiga, Bruna, and Pérez-Donoso 2023).”

Based on the scope of this work, it would be beneficial to include the effect of CuNPs in the viability of healthy cell lines.

Response. Done. We evaluated the cytotoxic effect of each Cu NPs over human oral epithelial cells (OKF6/TET2) (Dongari-Bagtzoglou and Kashleva 2006) after 1 and 3 h of exposition. Obtained results showed that after 1 h of exposition to NPs, oral cells do not show any variation compared with the control (absence of nanoparticles). We observed a negative effect with DMSO, decreasing the viability in 20%  (DMSO: control positive of detrimental agent over viability cell). After 3 h exposition a decrease in viability of oral cells was determined: ~ 6-8 % in presence of Cu0 NPs, ~ 31-43% in presence of CuO NPs, and ~ 11-16% in presence of Cu2O NPs. DMSO at 3 h exposition diminished the viability in 35-36%.

The list of abbreviations is incomplete, please revise for quorum sensing, autoinducers,  minimal inhibitory concentration, minimal bactericidal concentration, among others.

Response: Done. As suggested all abbreviations were checked and corrected in the new version of the ms. Lines 461-477

The authors are requested to revise the manuscript as per the suggestions provided above before further processing for consideration in this Journal.

Response: Done.

Comments on the Quality of English Language

Moderate editing of English language required

Response: Done.

Reviewer 3 Report

Comments and Suggestions for Authors

The manuscript entitled “Antimicrobial effect of Copper Nanoparticles on relevant supragingival oral bacteria” by Oetiker et al., investigates the potential of copper nanoparticles (Cu NPs) in dentistry for addressing bacterial dysbiosis and tooth decay. It analyzes the effects of three types of Cu NPs on the growth, viability, and biofilm formation of caries-associated and commensal oral bacteria. The authors claim that CuO nanoparticles (CuO NPs) demonstrate reduced toxicity towards commensal bacteria while exerting significant effects on pathogens, suggesting their potential for targeted treatments against oral pathogens. However, the absence of a positive control in the study raises concerns about the reliability of the results and their applicability in dentistry. Additionally, several other issues with the manuscript have been highlighted below:

- The study focuses solely on the in vitro effects of Cu NPs on bacterial growth and biofilm formation. While informative, extrapolating these findings to clinical settings may be challenging without further validation through in vivo studies.

- The authors should provide detailed characterization of the copper nanoparticles (Cu NPs) utilized in this study, employing techniques such as transmission electron microscopy (TEM), scanning electron microscopy (SEM), X-ray diffraction (XRD), and others.

-The selection of representative caries-associated and commensal oral bacteria may not fully capture the complexity of the oral microbiota. Including a broader spectrum of bacterial species could provide a more comprehensive understanding of the effects of Cu NPs.

- The study primarily assesses the outcomes of Cu NP exposure on bacterial growth and biofilm formation without delving into the underlying mechanisms. Understanding the mechanisms by which Cu NPs exert their effects on oral bacteria could enhance the interpretation and potential translatability of the findings.

- The authors must provide details regarding the synthesis of Cu NPs, specifying whether chemical methods or green synthesis were employed in their production process.

The authors are requested to revise the manuscript as per the suggestions provided above before further processing for consideration in this Journal.

Comments on the Quality of English Language

Moderate editing of English language required

Author Response

Comments and Suggestions for Authors

The manuscript entitled “Antimicrobial effect of Copper Nanoparticles on relevant supragingival oral bacteria” by Oetiker et al., investigates the potential of copper nanoparticles (Cu NPs) in dentistry for addressing bacterial dysbiosis and tooth decay. It analyzes the effects of three types of Cu NPs on the growth, viability, and biofilm formation of caries-associated and commensal oral bacteria. The authors claim that CuO nanoparticles (CuO NPs) demonstrate reduced toxicity towards commensal bacteria while exerting significant effects on pathogens, suggesting their potential for targeted treatments against oral pathogens. However, the absence of a positive control in the study raises concerns about the reliability of the results and their applicability in dentistry. Additionally, several other issues with the manuscript have been highlighted below:

The study focuses solely on the in vitro effects of Cu NPs on bacterial growth and biofilm formation. While informative, extrapolating these findings to clinical settings may be challenging without further validation through in vivo studies.

Response: We agree with the reviewer comment. The ms is a first contribution to understand the effect of Cu NPs over some relevant oral pathogens and commensal bacteria, and it’s not focused on the clinical application of it. With these in vitro models we aim to establish a foundation before transitioning to in vivo models.

In this context, other antimicrobial agents, applied either professionally or delivered from dentifrices or mouthwashes, were tested in vitro in a first stage. Using assays of bacterial growth and biofilm formation many compounds were demonstrated that control plaque formation or suppress cariogenic species. Our in vitromethodology enables the screening of diverse concentrations of various types of NPs. These findings are intended to identify the most effective copper NP for subsequent in vivo experiments.

We understand the limitations of this work and based on that we propose it as an initial study of the effect that different Cu NPs have on some oral relevant bacteria, and in our opinion represents a significant discovery with great clinical potential to be explored in the future.

- The authors should provide detailed characterization of the copper nanoparticles (Cu NPs) utilized in this study, employing techniques such as transmission electron microscopy (TEM), scanning electron microscopy (SEM), X-ray diffraction (XRD), and others.

Response: The nanoparticles used in this study are commercially available and were purchased from the company Nanotec https://nanotec.global/our-services/. Cu0, Cu2O, and CuO NPs with average sizes near 40 to 60 nm, were used in the experiments. These NPs have been previously used in several studies (patent products: WO2015035530A8 (Jarpa 2014). CL2013002853A1 (Jarpa 2013).

: 774111-5G Inventor. (Year). Title of patent (Country/Region Patent No. Number). Issuing Body. URL

Scanning electron microscopy (SEM) from chemically made Cu0 NPs.

Scanning electron microscopy (SEM) from chemically made Cu0 NPs.

Scanning electron microscopy (SEM) from chemically made CuO NPs.

-The selection of representative caries-associated and commensal oral bacteria may not fully capture the complexity of the oral microbiota. Including a broader spectrum of bacterial species could provide a more comprehensive understanding of the effects of Cu NPs.

Response: We appreciate the reviewer comment. However, as mentioned in the ms the aim of the present work was to specifically analyze the in vitro effect of Cu NPs on relevant cavity-associated bacteria (S. mutans and L. rhamnosus) as well as healthy oral-associated bacteria (S. sanguinis and S. salivarius).  In line 454 of our paper, we underscored the critical importance of analyzing the impact on oral samples to discern broader effects within the community.

Line 454: “Moving forward, our research aims to expand into in vivo studies to analyze the in vivo effect of these Cu NPs, due to pivotal importance to incorporating supragingival community to assess the competitive dynamics of Cu NPs. Overall, our findings underscore the potential of CuO NPs for targeted applications against oral pathogens, emphasizing their promise as a therapeutic strategy for enhancing oral health.”

We choose S. mutans and L. rhamnosus as cariogenic species, due to the impact of their biofilm formation on the dynamics of the supra gingival microbial community. In the context of caries, exposure of dental biofilms to dietary sugars leads to their fermentation into organic acids, resulting in a greater presence of acidogenic and aciduric species. S. mutans is particularly recognized for its capability to adhere to tooth surfaces, establishing biofilms that create an optimal environment for bacterial growth and acid production, consequently contributing to the demineralization process (Marsh 2003, 2010). Within this acidic environment, other acidogenic species, such as L. rhamnosus, can also thrive and form biofilms (Takahashi and Nyvad 2011) Hence, with acidogenic bacteria predominating, decrease the proliferation and biofilm formation of other species associated with oral health, such as S. sanguinis (Horiuchi et al. 2009).

For a better understanding of the strains tested, we included this information in the Background section.

- The study primarily assesses the outcomes of Cu NP exposure on bacterial growth and biofilm formation without delving into the underlying mechanisms. Understanding the mechanisms by which Cu NPs exert their effects on oral bacteria could enhance the interpretation and potential translatability of the findings.

Response: We agree with the reviewer, however this is not the focus of the present ms. We are currently studying the mechanisms associated with the effect of Cu NPs on cells of oral pathogens studied. We are focused on determining specific cellular targets and the production of reactive oxygen species that could explain their effect. We expect to publish these results at the end of 2024 or during 2025.

- The authors must provide details regarding the synthesis of Cu NPs, specifying whether chemical methods or green synthesis were employed in their production process.

Response: Done. The NPs were produced using chemical methods, however the specific production method is not publicly available by the selling company Nanotec. The size and composition of the NPs were indicated in the ms (see lines 146-148).

“Metallic Copper nanoparticles (Cu0, Cu2O and CuO) were obtained from NANOTEC S.A. (Santiago, Chile). Nanoparticles with size ranging ~ 30-60 nm were used in all the experiments (99,9978% purity). These NPs were made by chemical methods (https://nanotec.global/our-services/).”

Round 2

Reviewer 1 Report

Comments and Suggestions for Authors

The correction is done.

Author Response

Reviewer 1

Dear Authors, the study is interesting but some missing points you have.

  1. Please add in Introduction information about nanoparticles in medicine and environment and contemporary opinions about that. It should be also discussed in Discussion chapter.

Response: Done. As requested, the ms was modified (see new lines: 86-97).

“NPs has been used in medical and environmental applications. Nanotechnology revolutionizes the healthcare sector enhancing diagnosis accuracy, monitoring diseases, advancing surgical equipment, promoting regenerative medicine, refining vaccine development, and optimizing medication delivery systems. Moreover, it facilitates the development of cutting-edge research instruments, paving the path for the creation of innovative drugs to enhance treatments across a spectrum of ailments [17]. Also, nanotechnology holds immense promise in offering inventive solutions to diverse environmental challenges. These encompass enhanced pollution reduction techniques, advanced water treatment methods, precise environmental sensing technologies, efficient remediation processes, and the optimization of alternative energy sources to be more economically viable. Engineered nanomaterials possess distinctive properties that facilitate the development of these innovative technologies, paving the way for sustainable solutions to environmental issues [18].”

  1. Are there any other metal nanoparticles used in medicine/dentistry?

Response: Done. The information requested was incorporated in the ms (see new lines 83-84).

“In this context, different types of nanoparticles (NPs) have been applied in dental materials [15] (FeOx, ZrO2, Silica-based, TiO2 and Ag NPs) because play a pivotal role in dental applications, serving in dental fillings, enamel surface polishing to deter caries, and as implant materials surpassing conventional alternatives in efficacy [16].”

  1. Are there any products / materials already in dentistry with Cu?

Response: Done. The information requested was incorporated in the ms (see new lines 128-130).

In oral applications, it was reported that Cu NPs can be added to dental cements, restorative materials, adhesives, resins, irrigating solutions, obturations, orthodontic archwires and brackets, implant surface coatings, and the bone regeneration process [15–20].”

  1. The aim is already the conclusion; it should be redacted.

Response: Done. The ms was modified properly (see new lines 459-471).

The authors are requested to revise the manuscript as per the suggestions provided above before further processing for consideration in this Journal.

Response: Done.

Comments on the Quality of English Language

Moderate editing of English language required

Response: Done.

Haleem, Abid, Mohd Javaid, Ravi Pratap Singh, Shanay Rab, and Rajiv Suman. 2023. “Applications of Nanotechnology in Medical Field: A Brief Review.” Global Health Journal 7(2):70–77. doi: 10.1016/J.GLOHJ.2023.02.008.

Pathakoti, Kavitha, Manjunath Manubolu, and Huey Min Hwang. 2018. “Nanotechnology Applications for Environmental Industry.” Handbook of Nanomaterials for Industrial Applications 894–907. doi: 10.1016/B978-0-12-813351-4.00050-X.

Priyadarsini, Subhashree, Sumit Mukherjee, and Monalisa Mishra. 2018. “Nanoparticles Used in Dentistry: A Review.” Journal of Oral Biology and Craniofacial Research 8:58–67. doi: 10.1016/j.jobcr.2017.12.004.

Yun, Zhang, Du Qin, Fei Wei, and Li Xiaobing. 2022. “Application of Antibacterial Nanoparticles in Orthodontic Materials.” Nanotechnology Reviews 11(1):2433–50. doi: 10.1515/ntrev-2022-0137.

Reviewer 2 Report

Comments and Suggestions for Authors

The authors attended all suggestions

Author Response

Reviewer 2

Comments and Suggestions for Authors

The following comments can aid authors to improve the quality of manuscript:

In abstract, change antimicrobial to antibacterial.

Response: Done. See new line 31.

In introduction, define the acronym of NPs. In addition, give a brief overview of nanoparticles and their advantages as AOAs.

Response: Done. See new line 83 (NPs) and lines 98-102(Advantages as AOAs).

98-102

“The dentistry use of Nanotechnology not only garners patient interest for its potential cost-effectiveness and time-saving attributes but also offers psychological relief by minimizing treatment-related stress. The ongoing advancement of tailored nanomaterials holds promise in resolving dental issues. Although nanotechnology's current impact on oral disease treatment is somewhat constrained, ongoing research endeavors are poised to unlock significant advancements in the near future.[16,19]”

In introduction, explain the advantages of copper over other metal ions, synthesis methods, and what has been documented in the synthesis of nanomaterials for antibacterial applications.

Response: Done. See new lines 118-119 (advantages) and lines 104-111 (methods) of the ms.

118-119

“Besides, Cu presents advantages over other metals because of their abundant and relatively inexpensive, making Cu NPs more inexpensive for large-scale applications [19,25].”

104-111

“The synthesis of Cu-based nanoparticles can be by chemical or biological methods. The chemical way typically follows: "bottom-up" and "top-down" approaches. In the bottom-up method, atomic-level precursors are utilized to synthesize nanoscale materials. The top-down approach involves breaking down a bulk solid into progressively smaller components to obtain nanoparticles [21]. Furthermore, green synthesis routes have been utilized for both enzymatic and non-enzymatic production of Cu NPs. These methods involve the interaction of copper salt with organic compounds resulting in Cu NPs formation. This green way has several advantages including easy accessibility, non-toxicity, cost-effectiveness, and straightforward handling [22]. “

Please add the ATCC number of the cultured strains in this study.

Response: Done. Included in lines 150, 151 and 153.

“The Streptococcus genus bacteria (S. mutans ATCC 25175; S. salivarius ATCC 13419 and S. sanguinis SK36) were grown in Brain Heart Infusion broth (BHI)-bacitracin (0,2 units/mL). The solid medium utilized was based on tryptone, yeast extract, cystine (TYC) agar, bacitracin, and sucrose (5%). L. rhamnosus ATCC 53103 was grown in liquid De Man, Rogosa, and Sharpe (MRS) medium. The solid growth was MRS-Agar. All bacteria were grown at 37°C in microaerophilic conditions (candle jars).”

Explain what the positive control was used in this study and included in Figure 1 and 2.

Response: Done. The positive control of growth in absence of NPs was included in the Figures and the corresponding legends (lines 223 and 266). In addition, we included more details in materials and methods section (lines 165 and 172).

The explanation about the effect of the antibacterial activity of CuNPs can be associated with the physical and chemical data of these nanomaterials. It would be important to consult this with suppliers.

Response. We agree with the reviewer on this point. Accordingly, we discussed this possibility in the new version of the ms (see new introduction lines 120-126 and discussion lines 400, 411, 416).

“The toxicity mechanism of Cu NPs involving their interaction, accumulation, and subsequent dissolution within the cellular membrane, leading to alterations in membrane permeability. Also, the release of ions from the NPs induces the generation of reactive oxygen species (ROS), triggering lipid peroxidation, protein oxidation, and DNA degradation [26,27]. Furthermore, the metal ions present within the cell inflict damage upon DNA and interfere with ATP production. Cu ions, specifically, exhibit interactions with phosphate and thiol groups present in proteins and DNA, resulting in denaturation and other structural disturbances [23].”

Based on the scope of this work, it would be beneficial to include the effect of CuNPs in the viability of healthy cell lines.

Response. Done. We evaluated the cytotoxic effect of each Cu NPs over human oral epithelial cells (OKF6/TET2) (Dongari-Bagtzoglou and Kashleva 2006) after 1 and 3 h of exposition. Obtained results showed that after 1 h of exposition to NPs, oral cells do not show any variation compared with the control (absence of nanoparticles). We observed a negative effect with DMSO, decreasing the viability in 20%  (DMSO: control positive of detrimental agent over viability cell). After 3 h exposition a decrease in viability of oral cells was determined: ~ 6-8 % in presence of Cu0 NPs, ~ 31-43% in presence of CuO NPs, and ~ 11-16% in presence of Cu2O NPs. DMSO at 3 h exposition diminished the viability in 35-36%.

The list of abbreviations is incomplete, please revise for quorum sensing, autoinducers,  minimal inhibitory concentration, minimal bactericidal concentration, among others.

Response: Done. As suggested all abbreviations were checked and corrected in the new version of the ms. Lines 474-491

The authors are requested to revise the manuscript as per the suggestions provided above before further processing for consideration in this Journal.

Response: Done.

Comments on the Quality of English Language

Moderate editing of English language required

Response: Done.

Chatterjee, Arijit Kumar, Ruchira Chakraborty, and Tarakdas Basu. 2014. “Mechanism of Antibacterial Activity of Copper Nanoparticles.” Nanotechnology 25(13). doi: 10.1088/0957-4484/25/13/135101.

Dongari-Bagtzoglou, Anna, and Helena Kashleva. 2006. “Development of a Highly Reproducible Three-Dimensional Organotypic Model of the Oral Mucosa.” Nature Protocols 1(4):2012–18. doi: 10.1038/NPROT.2006.323.

Raffi, Muhammad, Saba Mehrwan, Tariq Mahmood Bhatti, Javed Iqbal Akhter, Abdul Hameed, Wasim Yawar, and M. Masood Ul Hasan. 2010. “Investigations into the Antibacterial Behavior of Copper Nanoparticles against Escherichia Coli.” Annals of Microbiology 60(1):75–80. doi: 10.1007/s13213-010-0015-6.

Ramos-Zúñiga, Javiera, Nicolás Bruna, and José M. Pérez-Donoso. 2023. “Toxicity Mechanisms of Copper Nanoparticles and Copper Surfaces on Bacterial Cells and Viruses.” International Journal of Molecular Sciences 24(13):10503. doi: 10.3390/ijms241310503.

Reviewer 3 Report

Comments and Suggestions for Authors

Authors have improved the manuscript but this study still lacking several drawbacks for instance:

-Authors should provide the TEM analysis of the used NPs,

-NPs size should be determined employing TEM experiment

-Comparison of the biological activity of the NPs must be compared with standard antimicrobial agents available commercially.

Just buying and testing the antimicrobial effect of NPs should not be considered for publications in highly reputed ISI journals with high impact factor.

Comments on the Quality of English Language

Minor editing of English language required

Author Response

Reviewer 3

-Authors should provide the TEM analysis of the used NPs, NPs size should be determined employing TEM experiment.

Response: Done. We agree with the reviewer. We believe that a more detailed information of the NPs must be included in the ms. To this end, we have incorporated additional details regarding the characteristics of the NPs, such as CAS number, molecular weight, size, etc. (Table 1, added in line 151 in new version of ms). Furthermore, transmission electron microscopy images depicting the size of the NPs utilized have been included to confirm the size of the NPs.

Table 1: Characteristics of the Cu NPs used in this work

Cu0 NPs

Cu2O NPs

CuO NPs

CAS number

7440-50-8

1317-38-0

1317-38-0

Molecular weight

 63.5 g/mol

143.9 g/mol

79.6 g/mol

Color

Brown-red

green

black

Size

~40-60 nm

~40-60 nm

~40-60 nm

Batch

180314-RN

230315-SP

190726-MA

A High-Resolution Scanning Electron Microscopy (INSPECT - F50 FEI) with detector Scanning transmission electron microscopy (STEM) and an electron acceleration of 10 kV was used to determine the size of the NPs.

11-5G Inventor. (Year). Title of patent (Country/Region Patent No. Number). Issuing Body.

Figure. Scanning transmission electron microscopy (STEM) analysis of A) Cu0, B) Cu2O and C) CuO NPs.

The STEM analysis determined spheric NPs with a size ranging from 40 to 60 nm. These NPs have been previously used in other studies (patents: WO2015035530A8, CL2013002853A1).

- Comparison of the biological activity of the NPs must be compared with standard antimicrobial agents available commercially.

Response: Done. Based on the reviewer's suggestion, a table showing the MICs values of some commonly used antimicrobial agents over S. mutans (cariogenic bacteria) and S. sanguinis (commensal bacteria) has been included in the new version of ms. Some standard AOAs and chitosan NPs were included in the Table S1; sodium fluoride, chlorhexidine, penicillin, chitosan, chitosan NPs, and daptomycin. Table I shows that most antimicrobial agents generate a more pronounced detrimental effect on S. sanguinis than S. mutans. However, some interesting exceptions occurs in the case of chlorhexidine (CHX) and penicillin, that results more toxic for S. mutans.

The ms was modified properly to include a discussion on the antimicrobial effect of standard AOAs over S. mutans and S. sanguinis (lines 395-403):

“Currently, several commonly used antimicrobial agents against pathogenic oral bacteria include sodium fluoride, chlorhexidine, penicillin, chitosan, and daptomycin. Interestingly, S. sanguinis exhibits greater sensitivity to most standard antimicrobial oral agents (AOAs) compared to S. mutans, contrasting with the situation observed with Cu NPs (Table S1). Only chlorhexidine (CHX) and penicillin appear to be more toxic for S. mutans. However, it has been documented that CHX-containing mouthwash can alter the salivary microbiome, leading to a more acidic environment and reducing nitrite availability in healthy individuals [12]. Regarding antibiotics such as penicillin, studies have reported alterations in the oral microbiota, which may subsequently affect the concentration of salivary antibodies [13]”.

Table S1. Effect of oral antimicrobial agents on S. mutans and S. sanguinis.

Antimicrobial agent

S. mutans (MIC µg/mL)

reference

S. sanguinis (MIC µg/mL)

reference

Sodium fluoride

625

 (Dong et al. 2012)

4

 (Qian, Zhang, and Xiao 1998)

Chlorhexidine

2.3

  (Dong et al. 2012)

25

 (Li et al. 2022)

Penicillin

0.05

  (Dong et al. 2012)

2

(Doern et al. 1996)

 Chitosan

1250

 (Aliasghari et al. 2016)

1250

 (Aliasghari et al. 2016)

Chitosan NPS

625

 (Aliasghari et al. 2016)

312

 (Aliasghari et al. 2016)

Daptomycin

18.7

 (Li et al. 2022)

10.7

 (Li et al. 2022)

Cu0 NPs

N.D. (>1000)

 This work

N.D. (>1000)

 This work

Cu2O NPs

500

 This work

1000

 This work

CuO NPs

400

 This work

800

 This work

Aliasghari, Azam, Mohammad Rabbani Khorasgani, Sedigheh Vaezifar, Fateh Rahimi, Habibollah Younesi, and Maryam Khoroushi. 2016. “Evaluation of Antibacterial Efficiency of Chitosan and Chitosan Nanoparticles on Cariogenic Streptococci: An in Vitro Study.” Iranian Journal of Microbiology 8(2):93.

Doern, Gary V., Mary Jane Ferraro, Angela B. Brueggemann, and Kathryn L. Ruoff. 1996. “Emergence of High Rates of Antimicrobial Resistance among Viridans Group Streptococci in the United States.” Antimicrobial Agents and Chemotherapy 40(4):891–94. doi: 10.1128/AAC.40.4.891.

Dong, Liping, Zhongchun Tong, Dake Linghu, Yuan Lin, Rui Tao, Jun Liu, Yu Tian, and Longxing Ni. 2012. “Effects of Sub-Minimum Inhibitory Concentrations of Antimicrobial Agents on Streptococcus Mutans Biofilm Formation.” International Journal of Antimicrobial Agents 39:390–95. doi: 10.1016/j.ijantimicag.2012.01.009.

Li, Xinwei, Yufei Wang, Xuelian Jiang, Yuhao Zeng, Xinran Zhao, Jumpei Washio, Nobuhiro Takahashi, and Linglin Zhang. 2022. “Investigation of Drug Resistance of Caries-Related Streptococci to Antimicrobial Peptide GH12.” Frontiers in Cellular and Infection Microbiology 12. doi: 10.3389/FCIMB.2022.991938/FULL.

Qian, W., J. Zhang, and X. Xiao. 1998. “[Research on Inhibition of Sodium Fluoride on Five Subgingival Bacteria in Vitro].” Hua Xi Kou Qiang Yi Xue Za Zhi = Huaxi Kouqiang Yixue Zazhi = West China Journal of Stomatology.
